# Frontoparietal network topology as a neural marker of musical perceptual abilities

M. Lumaca [1] ✉, P. E. Keller [1,2], G. Baggio [3], V. Pando-Naude [1],
C. J. Bajada [4], M. A. Martinez[5], J. H. Hansen[5], A. Ravignani[1,6], N. Joe [1], P. Vuust[1],
K. Vulić[7] & K. Sandberg [5]

Why are some individuals more musical than others? Neither cognitive testing nor classical localizationist neuroscience alone can provide a complete answer. Here, we test how the interplay of brain network organization and cognitive function delivers graded perceptual abilities in a distinctively human capacity. We analyze multimodal magnetic resonance imaging, cognitive, and behavioral data from 200+ participants, focusing on a canonical working memory network encompassing prefrontal and posterior parietal regions. Using graph theory, we examine structural and functional frontoparietal network organization in relation to assessments of musical aptitude and experience. Results reveal a positive correlation between perceptual abilities and the integration efficiency of key frontoparietal regions. The linkage between functional networks and musical abilities is mediated by working memory processes, whereas structural networks influence these abilities through sensory integration. Our work lays the foundation for future investigations into the neurobiological roots of individual differences in musicality.

A central goal in cognitive neuroscience is to determine how neuro-cognitive variability gives rise to interindividual differences in human cognitive capacities and behaviors[1–3]. In the past decade, this line of inquiry has gained traction in music science[4] with the quest to elucidate the biological bases of the spectrum of variability in musical aptitude within human populations[5]. This knowledge does not only facilitate the development of neuroimaging biomarkers useful in rehabilitation and educational settings[6,7], but also has the potential to inform us on the neurobiological underpinnings of musicality in humans[8]. Our study specifically focuses on musical competence, a universal and distinctly human perceptual ability[9] characterized by significant individual differences across the general population[10]. Musical perceptual abilities are deeply rooted into our neurobiology[9],

scaffolded by domain-general neurocognitive systems such as attention and working memory[11–13]. It remains a relatively unexplored issue how variability in the neural architecture of these systems affects the wide range of music perceptual competencies observed across individuals. Our research employs network neuroscience and graph theory on a comprehensive MRI, cognitive, and behavioral dataset ($n > 200$) to examine the neurocognitive markers of music perceptual abilities within the general population of music listeners.

Musical competence, the ability to perceive, remember, and discriminate music[14], is a critical component of human musicality and hinges on core perceptuo-cognitive skills, including sensory integration, auditory discrimination, attention, and working memory. As such, musical competence is underpinned by the activity of extensive

[1]Center for Music in the Brain, Department of Clinical Medicine, Health, Aarhus University & The Royal Academy of Music Aarhus/Aalborg, Aarhus, Denmark. [2]The MARCS Institute for Brain, Behaviour and Development, Western Sydney University, Penrith, Australia. [3]Language Acquisition and Language Processing Lab, Norwegian University of Science and Technology, Trondheim, Norway. [4]Department of Physiology and Biochemistry, Faculty of Medicine and Surgery, University of Malta / University of Malta Magnetic Resonance Imaging Research Platform, Msida, Malta. [5]Center of Functionally Integrative Neuroscience, Department of Clinical Medicine, Health, Aarhus University, Aarhus, Denmark. [6]Department of Human Neurosciences, Sapienza University of Rome, Rome, Italy. [7]Department for Human Neuroscience, Institute for Medical Research, University of Belgrade, Belgrade, Serbia.
✉e-mail: massimo.lumaca@clin.au.dk

sensory and cognitive networks[15]. As a universal human ability emerging early in human development[16–18], it differs from music production, which necessitates specialized motor skills and technical musical knowledge (with the partial exception of singing[19,20]). It displays significant variability within the general population, especially in the pitch and rhythmic domains[10,21]. The neural investigation of perceptual abilities that are minimally dependent on formal musical training may offer deeper insights into the biological bases of individual differences in musical abilities[5]. Despite being grounded in the activity of complex networks, musical abilities have been mainly addressed in neuroscience either focusing on the neural activation[22,23] and morphology[24,25] of single brain regions or on the pairwise associations (functional[26,27] or structural[28–31]) of specific brain areas. A more holistic and richer understanding of the neural underpinnings of musical abilities would benefit from a network-level approach on brain data[32] in combination with an objective test of music perception.

Network neuroscience, recognizing the distributed nature of music perceptual faculty, offers a compelling framework for this endeavor[33,34]. It conceptualizes the brain as a complex network[35]: a wiring diagram (or "graph") consisting of nodes (distinct brain regions or neural populations) linked by edges. In functional brain networks, edges represent temporal relationships between activity in different brain regions (e.g., based on estimates of functional connectivity[36]). In structural brain networks, edges represent white matter tracts that connect these regions, most often obtained using diffusion tractography[37]. Network representations of the human brain offer two main benefits. Firstly, they facilitate the description of higher-order multivariate connectivity patterns and are thus more informative than simpler estimates of bivariate connectivity. Even relatively basic cognitive processes rely on the activity of interconnected elements within a networked system. Secondly, network representations allow for quantitative analysis of functional and structural connectivity patterns within a unified mathematical framework: graph theory[38]. Leveraging graph theory, network neuroscience quantifies three key properties of brain networks: *integration* (enabling efficient processing of distributed information), *segregation* (supporting specialized processing within localized clusters), and *centrality* (highlighting the importance of hub regions in functional integration)[39]. By employing graph theory on both structural and functional brain networks, this approach effectively bridges the domains of neuroanatomy and brain dynamics. The static architecture of functional and structural brain networks may thus provide the basis to understand the links between anatomical organization, information processing, mental representations, and behavior[40,41]. Research studying functional brain networks from resting-state fMRI (rs-fMRI) and structural brain networks from diffusion MRI (dMRI) indicates that nuances in such network properties relate to differences in general cognitive abilities, including intelligence, WM capacity, and cognitive control[42–47]. Such findings highlight the potential of network neuroscience to inform our understanding of how variation in relatively static aspects of brain organization shapes variability in human cognition and behavior, including music perception and production.

Standardized objective tests of music perceptual abilities include the Musical Ear Test (MET)[48], the Advanced Measures of Music Audiation[49], the Profile of Music Perception Skills[50], and the Swedish Musical Discrimination Test[51]. These tests mainly assess abilities in detecting pitch and timing variations in musical sequences, providing reliable measures of music competence that can be effectively related to brain network configurations. Among these tests, the MET, a well-established test of musical aptitude, has undergone validation in large-scale studies with listeners from different countries and diverse cultural backgrounds[21,52–54]. It is notable for its openly accessible format, correlating robustly with musical imitation scores used in musical academies, without being influenced by demographic factors such as age, sex, socio-economic status[52], or personality traits[21]. Despite its

short duration (~20 minutes) compared to the other tests (>40 minutes), the MET maintains robust psychometric properties[48]. Unlike aptitude or talent, 'competence' is a term that remains neutral, not favoring either innate qualities (nature) or learned skills (nurture). This test is effective in assessing interindividual differences in individuals without formal musical training ('non-musicians'), accounting for latent musical capacities that can lead these individuals to outperform the average musician's MET scores[21]. The MET is also positively correlated with implicit (self-reported) measures of general musical sophistication, like the widely used Goldsmiths Musical Sophistication Index[21,55]. Consequently, the MET, when used in combination with network neuroscience, is a reliable and ecologically valid tool for exploring the network-level neural foundations of the human capacity for music and in assessing how subtle neural variations affect musical skills in the general population.

The biological roots of the human capacity for music have garnered increased attention in the past decade within the fields of cognitive science and biomusicology[8,56]. Despite limitations in defining the components of musicality, this faculty is thought to arise from the complex synergy of various perceptuo-cognitive elements, each with unique neurobiological foundations and evolutionary histories[56]. Working memory (WM), a neurocognitive system with limited capacity crucial for temporary storage and manipulation of sensory information[57], is a foundational component of musical perceptual abilities[12] with a well-characterized network topology[58–62]. Research in music psychology and neuroscience indicates a hierarchical process in music perception involving serial-to-parallel conversion, integrating auditory elements into increasingly complex musical structures, from basic chunks to complete melodies[63]. Such a process hinges on the WM system's ability to retain lower-level units while integrating new information to form more elaborate musical constructs. Musical abilities, as assessed with the MET, correlate significantly with WM capacity: greater WM capacity often translates to superior musical skills[14,64]. The brain network underpinnings of this relationship remain unexplored. WM relies on the integration of past and current sensory information from a large-scale network[65], whose core infrastructure comprises bilateral dorsolateral prefrontal cortices and posterior parietal cortices[66–68]. Higher WM scores are associated with strongly integrative networks, promoting efficient inter-regional communication and rapid combination of information from distributed regions[65,69–71]. Previous research showed that children with musical training exhibit improved cognitive flexibility compared to their non-trained counterparts, a phenomenon linked to increased brain activations in frontoparietal regions[11]. The extensive characterization of frontoparietal networks and their relevance to musical behaviors allow us to formulate strong a priori hypotheses regarding their potential to serve as brain network markers for music perception. Exploring the relationship between MET scores, WM, and frontoparietal network topology, both at a functional and structural level, could enhance our understanding of the impact of domain-general neurocognitive systems on music perception and how they help shaping individual differences[72].

Our study employs graph theory analyzes to investigate the impact of functional and structural WM neural organization on music competence, integrating multimodal neuroimaging with behavioral and cognitive data. While previous research has focused on the effects of music listening and training on brain network configuration[73–77], our investigation reverses this viewpoint. We specifically explore how subtle interindividual differences in the frontoparietal network (FPN) relate to differences in music competence, assessed using the MET. Our hypothesis is that more globally efficient functional and structural FPNs may enhance the hierarchical processing and integration of musical elements, their retention in a temporary buffer, and their comparison with previous musical information, thereby positively impacting the musical competence of listeners. To assess the cognitive

capabilities of our participants, we collected from them cognitive data using the Wechsler Adult Intelligence Scale, Fourth Edition (WAIS-IV)[78], which includes a WM index. Musical ability extends beyond mere perception and encompasses emotional and affective components contributing to musical experience[79]. Alongside cognitive and behavioral assessments, participants completed the Goldsmiths Musical Sophistication Index (Gold-MSI)[10], a self-report questionnaire detailing formal and informal musical behaviors, experiences, and skills, which includes an evaluation of affective reactions to music. This addition allows us to explore whether the positive relationship between affective and perceptual capacities reported in previous studies[10] is partly rooted in the neural organization of WM systems. Neuroimaging research on individual differences requires large sample sizes ($n > 100$) to obtain reproducible findings[80–82]. Also, graph theory should be applied to both functional and structural modalities to achieve a comprehensive understanding of how the organization of functional and structural brain networks differently contributes to human behavior. The large sample size of this study, in combination with a network-of-interest approach that relies on a priori hypotheses, aims to address the issue of low statistical power in individual differences studies. Our use of graph theory in both functional and structural brain networks can further help in assessing their distinctive role in support of human behavior. The anticipated results are expected to shed light on the impact of brain network variability on the broad spectrum of musical abilities observed in humans and to inform future theories of how biological variability may impact music diversity within and across human cultures[83–86].

## Results

### Overview of the experimental design and analysis pipeline

Our dataset comprised MET[48], Goldsmiths Musical Sophistication Index (Gold-MSI)[10], and Wechsler Adult Intelligence Scale (WAIS-IV)[78] scores, along with functional and structural scans, from a large number of healthy adults (Table 1). Participants were mostly non-musicians (Suppl. Fig. 1). Non-musicians were selected in order to isolate neurobiological factors related to natural musical perceptual abilities, minimizing the influence of well-documented environmental factors, such as exposure to music and years of music training[87]. Our study aimed to elucidate the interplay between the structural and functional architecture of the frontoparietal network (FPN) and musical competence. We assessed the relationship between individuals' propensities

for integration or segregation within the FPN and their musical ability. This entailed examining how musical competence correlates with the organization of FPN nodes, either in facilitating efficient communication and integration across distal nodes (indicated by higher global efficiency and centrality) or in forming specialized, segregated clusters (reflected by higher clustering coefficient and local efficiency). Global efficiency is the most commonly used measure of functional integration, while in structural brain networks efficiency and centrality are the relevant metrics[88]. Additionally, we probed the potential mediating role of a domain-general cognitive feature, namely WM, in these brain-behavior correlations. Musical competence was quantified using the percentage of MET total scores.

The analysis pipeline of this study is depicted in Fig. 1 (see "Methods" for a detailed description). For each participant, an anatomical image underwent parcellation using the Destrieux atlas[89,90], which subdivides the human cerebral cortex into 148 distinct sulcal-gyral cortical parcels (74 homologous regions) based on anatomical landmarks and cytoarchitectonic boundaries. The resulting parcellation was fed into the pipeline for the construction of functional and structural networks (or connectivity matrices). In connectivity matrices, each parcel is a node, and each pair of nodes is connected via a functional or structural link (edge). These connectivity matrices display a comprehensive map of connections between all pairs of nodes in the parcellated brain. From these connectivity matrices, 16 cortical nodes were selected to define two new subnetworks: a frontoparietal (FPN) network of interest and an occipital network which served as a control. Figure 2 displays these two subnetworks. The FPN, selected for the main analysis, included the bilateral dorsolateral prefrontal cortex (middle and superior frontal gyrus and sulcus)[91,92] and bilateral posterior parietal cortex (inferior and superior parietal lobule, and intraparietal sulcus)[93,94]. In separate analyzes, functional and structural matrices were filtered using a multi-threshold approach and were binarized to create adjacency matrices. Graph theory metrics of global efficiency, local efficiency, clustering coefficient, and betweenness centrality[88] were calculated from these subnetworks (Fig. 3) and entered as dependent variables in a multiple regression analysis where the percentage of MET total scores was the main predictor. A mediation analysis using structural equation modeling (SEM) assessed the indirect influence of WM on the effect of topological nodal metrics on the percentage of MET total scores.

### Graph theory results for the structural networks

Figure 4A shows the results for the structural FPN (Suppl. Table 5). We observed a positive correlation between percentage of MET total scores and centrality in the right superior frontal gyrus (SupFG) (F = 4.40, pFDR = 0.0002) and the right superior parietal lobule (SupPL) (F = 3.70, pFDR = 0.002). Additionally, in the right SupPL, percentage of MET total scores was associated positively with global efficiency (F = 3.16, pFDR = 0.029). In contrast, a negative association was found between percentage of MET total scores with segregation measures in the right SupFG (local efficiency: F = −4.34, pFDR = 0.0003; clustering coefficient: F = −4.78, pFDR = 0.00004) and SupPL (local efficiency: F = −4.21, pFDR = 0.0003; clustering coefficient: F = −4.61, pFDR =0.00006). These results suggest that individuals exhibit superior musical perceptual abilities when the topology of FPN physical pathways, within these two brain regions, implies stronger potential for functional integration. Conversely, a potential for functional segregation in these two nodes is associated with comparatively worse musical perceptual abilities. No significant results were found between graph theory metrics in the occipital control network and MET scores (Suppl. Table 6).

### Graph theory results for the functional networks

Figure 4B displays the results for the functional FPN (Suppl. Table 7). A positive correlation was found between the global efficiency of the

**Table 1 | Descriptive statistics for MET and Gold-MSI subscales (N = 241)**

| | Mean | SD | Range |
|---|---|---|---|
| MET | | | Score Range |
| Total | 73.53 | 9.16 | 38–94 |
| Melody | 35.99 | 5.52 | 24–49 |
| Rhythm | 37.80 | 4.59 | 25–48 |
| Gold-MSI scores | | | Score Range (theoretical max) |
| Active engagement | 30.32 | 9.17 | 25–48 (63) |
| Perceptual abilities | 41.95 | 8.09 | 17–63 (63) |
| Musical training | 14.04 | 9.28 | 2–47 (49) |
| Emotions | 28.97 | 6.67 | 9–43 (49) |
| Singing abilities | 24.10 | 7.46 | 9–49 (42) |
| General sophistication | 59.61 | 16.90 | 27–121 (126) |

MET: Melody and Rhythm scores were calculated from 52 trials. Total scores were calculated from 104 trials. In this table, we report the descriptive statistics of the raw scores. Gold-MSI: Active Engagement (amount of time and money spent in music-related activities); Perceptual Abilities (self-reported accuracy of music listening skills); Musical Training (amount of formal musical training received); Emotions (ability to talk about emotions expressed in music); Singing (self-reported accuracy of one's own singing); General Sophistication (general index of musical sophistication).

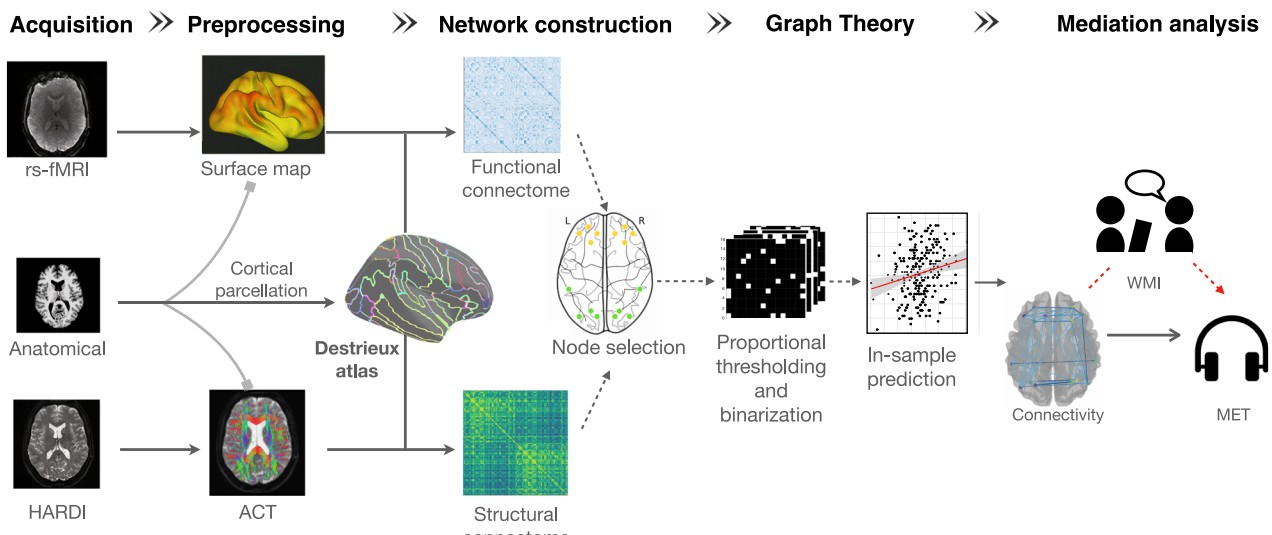

**Fig. 1 | Flowchart of the main analysis pipeline for single-participant data.** Anatomical data is parcellated into 148 cortical parcels (nodes) using the Destrieux parcellation scheme. The fMRI surface data is aligned to this parcellation using functional-structural coregistration, and spontaneous BOLD activity is extracted from each node. Statistical relationships of this activity are computed for each pair of nodes, creating a functional connectivity matrix (functional connectome; in blue) that highlights the functional connections between all the cortical regions (148 × 148). In parallel, whole-brain ACT is performed for HARDI images and aligned to the Destrieux's atlas, obtaining pairwise white matter associations between all nodes (structural connectome; in green) (148 × 148). After defining two specific subnetworks of 16 nodes each−frontoparietal and occipital−graph theory and multiple regression analyzes are applied to examine the relationship between musical scores and network topology (16 × 16). Finally, a mediation analysis is conducted to determine if the relationship between brain network topology and music perceptual scores is mediated by working memory capacities. Abbreviations: ACT anatomically constrained tractography, HARDI High Angular Resolution Diffusion Imaging, rs-fMRI resting-state functional MRI, MET musical ear test, WMI Working Memory Index.

right middle frontal gyrus (MFG) and percentage of MET total scores (F = 3.06, pFDR = 0.043), indicating that in individuals with higher music perceptual abilities, the right middle frontal gyrus efficiently communicates with and most likely integrates specialized information from other FPN regions. Notably, no significant correlations emerged between the graph theory metrics of the occipital control network and MET scores (Suppl. Table 8).

### Relationship between WM, music competence, and FPN topology

A significant relationship (r = 0.21, pFDR <0.01) was observed between WMI scores from the WAIS-IV and percentage of MET total scores (Suppl. Fig. 3), indicating that higher WM scores are linked to superior musical perceptual abilities in the MET. To explore the association between FPN topology and WM performance, we examined the impact of graph theory metrics (dependent variable), alongside WMI scores (independent variable), age, sex, and Musical Training Index (nuisance regressors), in a linear multiple regression framework. Our analysis of functional brain networks revealed a positive correlation between WMI scores and the global efficiency of the right MFG (F = 3.18, r = 0.20, pFDR =0.02), supramarginal gyrus (F = 2.70, r = 0.19, pFDR = 0.03), and superior frontal sulcus (F = 2.71, r = 0.17, pFDR = 0.03) (Suppl. Table 9). This result suggests that efficient communication and integration capabilities of these right hemispheric brain regions within the FPN are conducive to enhanced WM performance. Conversely, no significant associations were observed with WMI scores using nodal metrics of the functional occipital control network or metrics from structural networks (FPN and occipital) (Suppl. Tables 10–12).

### WM mediates the relationship between rMFG efficiency and music competence

A mediation model was developed to explore the relationship among the efficiency of the right MFG (rMFG) within the functional FPN, WM performance, and music competence. We hypothesized that WM would mediate the relationship between rMFG efficiency and musical competence. To test this hypothesis, we constructed a structural equation model (SEM). The model included rMFG global efficiency as predictor and the percentage MET total scores as dependent variable. Bootstrap resampling procedures were leveraged to derive robust, bias-corrected, accelerated confidence intervals for the parameters of interest. Table 2 and Fig. 5 display the statistics of mediation effects in the SEM. The SEM indicated a significant direct impact of rMFG efficiency on WMI scores (standardized beta = 0.21, p = 0.002). The direct path from WMI to percentage of MET total scores was also significant (standardized beta = 0.18, p = 0.010). Additionally, rMFG efficiency had a significant direct effect on musicality (standardized beta = 0.17, p = 0.013). Critically, the indirect effect of rMFG efficiency on musicality, mediated through WMI, was significant (standardized beta = 0.21, 95% CI [1.180 to 12.477]). In summary, the results demonstrate both direct effects of global neural efficiency on WM and musicality as well as an indirect pathway linking neural function to musical competence through domain-general cognitive abilities.

### Relationship between emotion, musical competence, WM and FPN topology

A significant correlation was found between Gold-MSI Emotions subscale and musical competence (Suppl. Fig. 3). This suggests that skills in expressing musical emotions verbally affect, or are affected by, musical abilities. However, no significant correlation was found between Gold-MSI Emotions subscale and WMI (Suppl. Fig. 3; r = −0.02, pFDR = 0.74). Similarly, no significant associations were observed between the Emotions subscale and nodal metrics of the functional and structural networks (FPN and occipital) (Suppl. Tables 13–16).

## Discussion

Using graph theory to analyze a large dataset of over 200 individual brains (diffusion and resting-state fMRI), combined with cognitive and musical ability assessments, we show how domain-general frontoparietal networks (FPNs) implicated in working memory (WM) influence

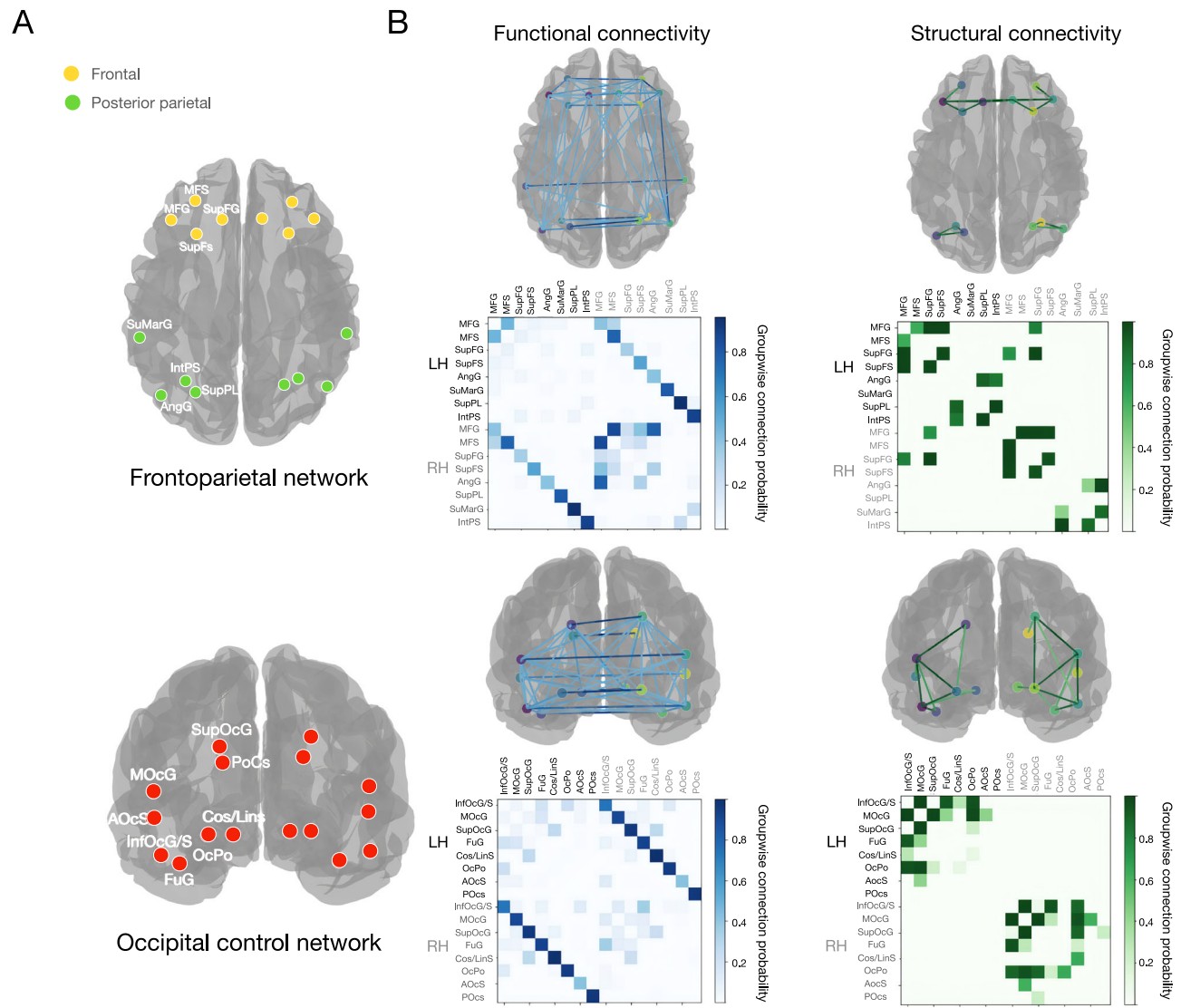

**Fig. 2 | Target networks for graph theory analysis. A** The frontoparietal network included frontal cortical nodes (yellow) subserving high-order cognition and posterior parietal nodes (green) involved in sensory integration. The occipital network we used as a control consisted of occipital, occipito-temporal and occipito-parietal cortical nodes supporting visual information processing. Each network contained eight homologous nodes per hemisphere. **B** Diagrams for functional (left) and structural brain networks (right). The color intensity of each edge reflects the proportion of participants exhibiting a connection between two nodes (range 0:1) across multiple network thresholds. The graph only displays the top 50% of the most robust connections. Darker blue signifies connections present in all or nearly all participants, a consistency maintained across all proportional thresholds from 0.15 to 0.30 in 0.01 increments. Complementing heatmaps are shown below each diagram. Abbreviations: LH - Left Hemisphere; RH - Right Hemisphere; MFG - middle frontal gyrus; SupFG - superior frontal gyrus; MFS - middle frontal sulcus; SupFS - superior frontal sulcus; SuMarG - supramarginal gyrus; IntPS - intraparietal sulcus; SupPL - superior parietal lobule; AngG - angular gyrus; InfOcG/S - inferior occipital gyrus (O1) and sulcus; MOcG - middle occipital gyrus (O2, lateral occipital gyrus); SupOcG - superior occipital gyrus (O1); FuG - lateral occipito-temporal gyrus (fusiform gyrus, O4-T4); Cos/Lins - medial occipito-temporal sulcus and lingual sulcus; OcPo - occipital pole; AoCs - anterior occipital sulcus and preoccipital notch (temporo-occipital incisure); PoCs - parieto-occipital sulcus (or fissure).

music perception skills in humans. Specifically, we found that higher global communication efficiency in structural and functional FPNs, notably in key areas of the right dorsolateral prefrontal cortex and the right superior parietal lobule, correlated positively with enhanced music perceptual skills. Critically, differences emerged when comparing results for functional and structural networks. For functional networks, global efficiency in right middle frontal gyrus (rMFG) was significantly associated with both musical competence and WM, the latter serving as a mediator in the direct influence of FPN organization on musical aptitude. Conversely, such direct association with WM performance was not observed for key regions of the structural networks. These results were based on brain images acquired during rs-fMRI and diffusion MRI scans, not during a musical or cognitive task.

The findings suggest that the inherent organization of FPN's core components may serve as a neural marker for musical perceptual abilities in the general population, with distinct roles played by functional and structural network organization.

Musicality, a multifaceted human trait, is the product of domain-specific perceptual skills, such as pitch and rhythm perception, and broader cognitive functions like attention and WM. Research into its neurobiological roots necessitates a divide-and-conquer approach, deconstructing musicality into its fundamental perceptual and cognitive components for isolated examination[56,95]. Previous research has mainly focused on the anatomy and function of auditory networks to elucidate the neurobiological bases of music perceptual components and their variability. These findings show that the diverse performance

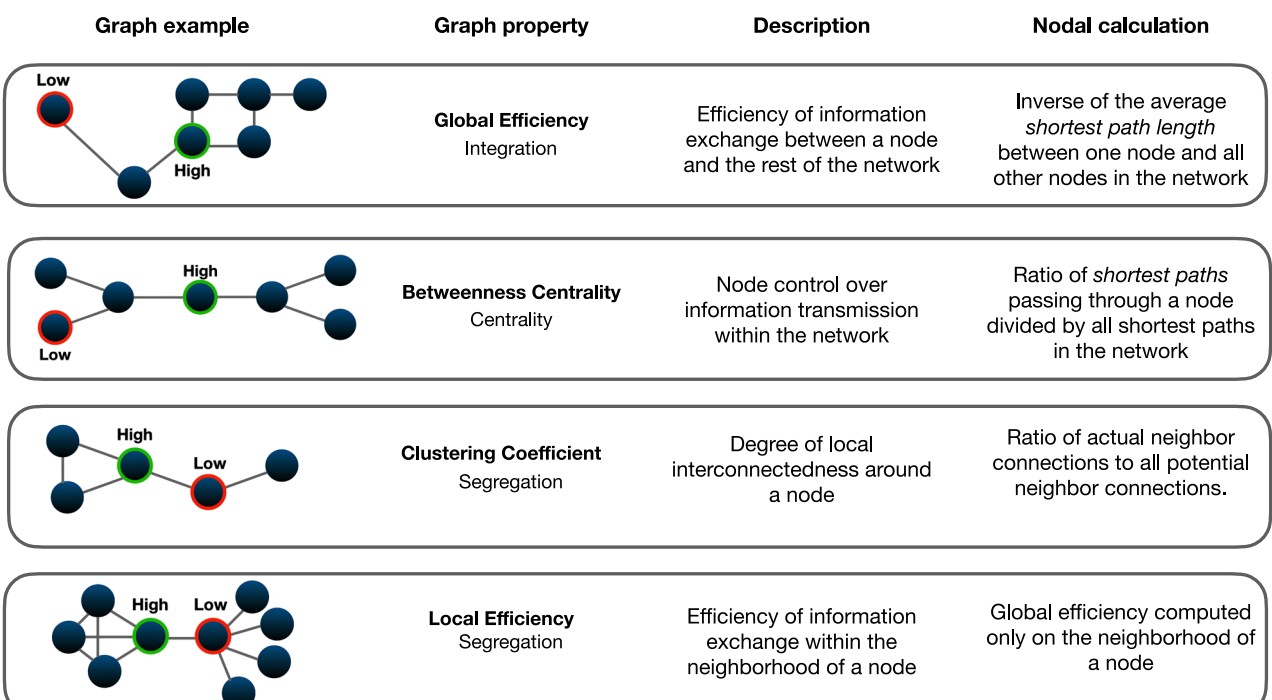

**Fig. 3 | Schematic representation of the four graph metrics computed for each node of the network.** These metrics reflect three key informational properties of brain networks: *integration* (global efficiency), *centrality* (betweenness centrality), and *segregation* (clustering coefficient and local efficiency). In the diagram, nodes where the graph properties register low values are marked with red circles, whereas nodes exhibiting high values are denoted with green circles. Shortest path length is defined as the minimum number of steps required to travel from one node to another within the network. Higher shortest path length values suggest that more steps are needed to connect two nodes with adjacent nodes (or neighbors) having a path length of 1. A node has high global efficiency if it can quickly reach other nodes through short paths, making it efficient in spreading information across the network. A node displays high betweenness centrality if it is crossed by a large number of shortest paths in the network. A high clustering coefficient is found in nodes with neighbors that are highly interconnected. A similar measure is the one of local efficiency. A node has high local efficiency if the average shortest path length among its neighboring nodes is low (i.e., its neighbors can efficiently exchange information among themselves).

in discriminating and encoding music can be predicted by differences in the spontaneous activity of the auditory cortex[23] and in how strongly the left and right auditory cortices are functionally connected[26,27,96]. Additionally, greater integrity of cross-callosal auditory projections is associated with superior music-related perceptual skills[30,97]. So far, the examination of the neural foundations of musical aptitudes has faced two main constraints: relatively small sample sizes ($n = 20$–$100$) reducing result replicability[80,81] and a narrow focus on pairwise connectivity between brain regions. These approaches have been effective in highlighting the key role of bilateral auditory networks in music perceptual abilities. However, they neglect the myriad of other connections that constitute the human brain architecture and that may support other components of musicality. Network-based methods for studying brain connectivity are optimal for gaining insights into how clusters of brain regions coordinate to support cognitive and perceptual capacities effectively.

A pivotal discovery of this study is the significant role played by the functional topology of the rMFG, a key region of the dorsolateral prefrontal cortex (DLPFC), in predicting music perceptual abilities in a large sample of individuals. We found that this effect was partly mediated by WM. The rMFG is critical for higher-level cognitive processes, including executive functions and WM[98–100]. Accordingly, functional studies linking WM and music perception often report DLPFC activity[101,102]. Platel and colleagues[103] observed bilateral activation in Brodmann areas 9 and 10 of the middle frontal gyri during an episodic music memory task, using PET scans. They attributed this DLPFC activity to the perceptual analysis of melodies in WM. Notably, the rMFG is also implicated in the perception of rhythmic structure in music[104,105], underscoring the critical role of WM functions in the

perception of durations in auditory stimuli. One hypothesis is that DLPFC might mediate the critical memory processes required during music perception (encoding, maintenance, and integration) via interaction with posterior parietal regions, likely the seat of internal sensory or perceptual representations[58]. Our findings indicate that this operation is more effective when an efficient functional organization is in place, such as more direct functional routes between rMFG and the rest of the network. This would explain the superior music abilities in participants with higher intrinsic global efficiency in the prefrontal region. Our finding corroborates network neuroscience research indicating that a network's information processing performance can be augmented by sparse functional configurations that yield disproportionately high efficiency[43,106]. Here, we show that this type of organization in FPNs can provide a neuromarker of music competence in the general population.

The lack of a direct one-to-one correspondence between structural and functional networks[107] may account for the divergent outcomes observed in our graph-theoretic analyzes of these networks. We observed that the integration capabilities and centrality of the right superior parietal lobule within the structural FPN network predict musical competence but do not correlate with WM performance. This finding is consistent with prior research indicating a stronger association of cognitive performance with functional rather than structural connectivity[108–111]. Resting-state functional connectivity is more closely associated with high-level cognitive tasks such as WM likely due to its dynamic and flexible nature[112], which aligns with the complex and temporally integrated demands required by these tasks. Conversely, structural connectivity is more closely linked to tasks with significant sensory components, such as spoken language perception[111,113],

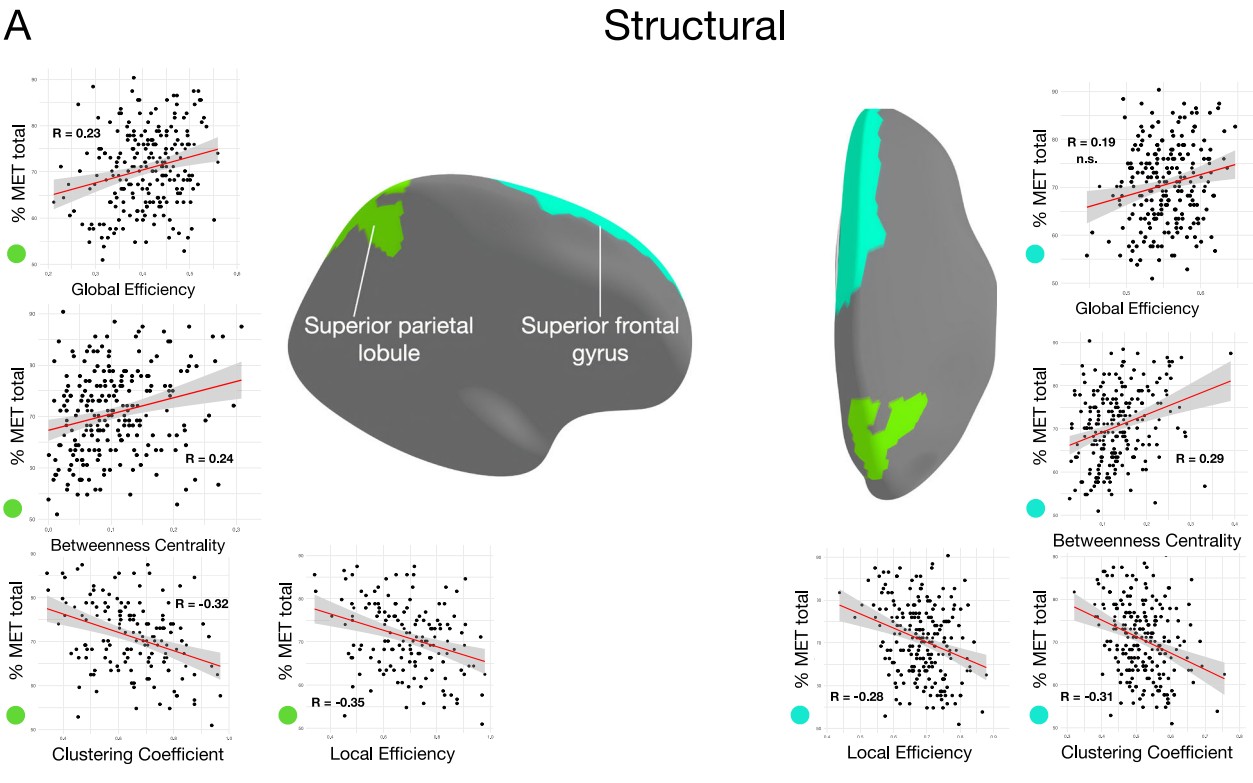

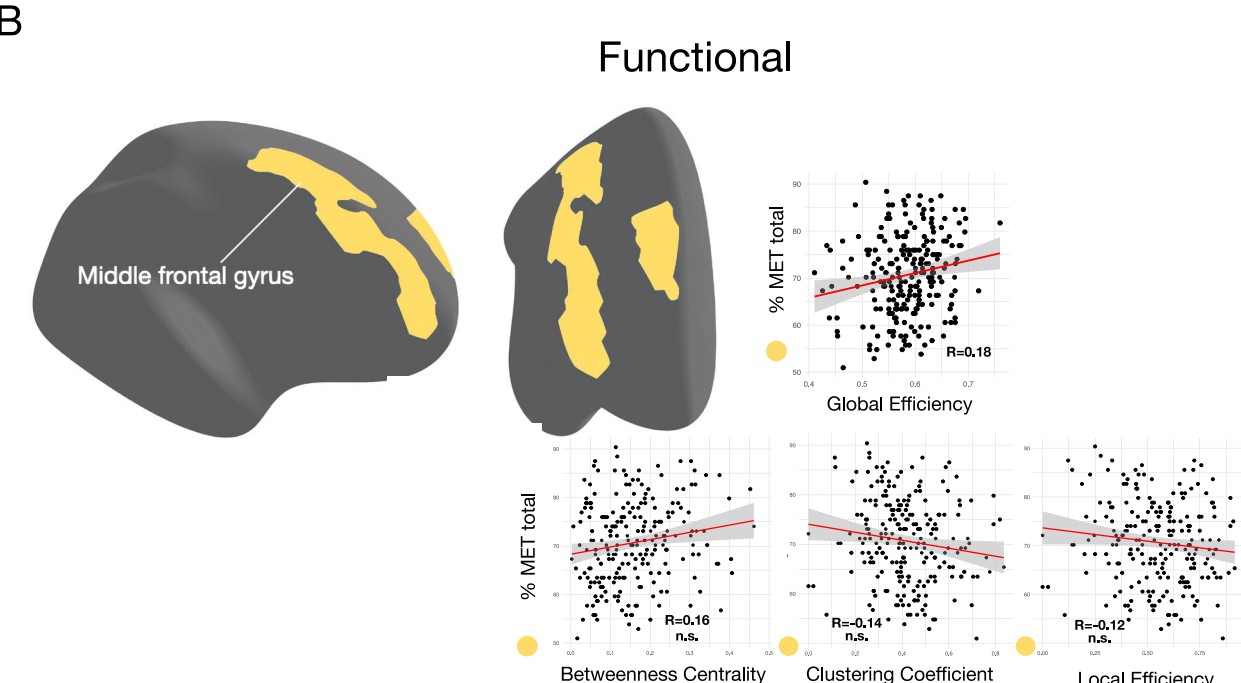

**Fig. 4 | Neurobehavioural correlations for the structural and functional fron-toparietal networks. A** Scatterplots showing linear relationships between graph theory metrics of *integration* (global efficiency), *segregation* (clustering coefficient and local efficiency) and *centrality* (betweenness centrality) (x-axis) and percentage of MET total scores (% MET total; y-axis) for the right superior parietal lobule (light green color; on the left) and the right superior frontal gyrus (cyan color; on the right) within the frontoparietal network (*N* = 225). **B** Scatterplots for the association between graph theory metrics for the right middle frontal gyrus (rMFG) within the functional frontoparietal network (yellow color) and percentage of MET total

scores (*N* = 232). Only the relationship with the global efficiency of this node was significant. Multiple regression analysis using GLM was conducted to assess the statistical significance of the correlations shown in both panels. For each scatter-plot, the Pearson's correlation coefficient (R), along with the corresponding F-values and false discovery rate (FDR)-corrected *p*-values (q < 0.05, two-tailed), are reported in the Results section. The shaded gray area in the scatterplots represents the 95% confidence interval around the regression line. Each data point on the scatterplots corresponds to an individual participant.

**Table 2 | Results from the structural equation model (SEM)**

| Path | Nodes connected | Est. β | SE | 95% CI | *p*-value |
|------|----------------|--------|-----|--------|-----------|
| c | Global E rMFG -> WMI | 42.265 | 15.014 | [17.981, 75837] | 0.002 |
| a | WMI - > MET | 0.115 | 0.044 | [0.011, 0.197] | 0.010 |
| b | Global E rMFG -> MET | 24.518 | 9.913 | [5.056, 44.023] | 0.013 |
| Indirect | c*a | 5.335 | 2.028 | [1.180, 12.477] | 0.043 |
| Total | c*a + b | 29.853 | 9.654 | [11.434, 47.269] | 0.002 |

The mediation analysis was performed using the lavaan package in R. The SEM model was specified to examine the relationships between the neural metric, working memory subscale of WAIS, and musicality. The model was fitted using the maximum likelihood estimation method with bootstrapped standard errors (1000 iterations). Standardized path coefficients and their corresponding *p*-values (two-tailed) are reported above. Abbreviations: Global E rMFG (Global Efficiency for rMFG in the functional network); WMI (Working Memory Index); MET (percentage of MET total score).

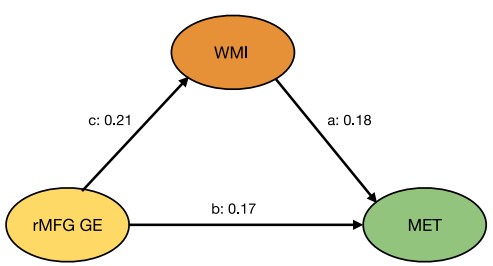

**Fig. 5 | Path diagram for the mediation analysis using structural equation modeling (SEM).** Standardized coefficients are shown for each path. The bootstrap statistical significance of the direct and indirect paths is presented in Table 2. Results of the proposed model confirm that the global efficiency of the right middle frontal gyrus within the functional frontoparietal network is positively associated with percentage of MET total scores, through greater working memory abilities. Abbreviations: rMFG - right middle frontal gyrus; GE - global efficiency; MET - percentage of MET total scores; WMI - Working Memory Index.

reflecting its role in establishing stable physical pathways for sensory information processing. In our study, we extend this finding to musical perceptual abilities. The superior parietal lobule, recognized as a higher-order association area and an integrative hub[114,115] is thought to encode and combine past and current sensory information, influencing integrated representations for guiding subsequent adaptive behavior. An alternative, not mutually exclusive, interpretation is that the right superior parietal lobule is implicated in sensory-related attentional processes essential for music perception. This region, together with the superior frontal gyrus, is a key area of the dorsal attention network[116]. Activity in this dorsal FPN reflects active goal-directed control of attention[117]. WM is thought to encompass two distinct operations with different neuroanatomical locations: firstly, a selection mechanism that retrieves pertinent items, and secondly, an updating function that redirects attentional focus[118]. This updating process is characterized by transient activation in the superior frontal and posterior parietal cortices. This observation aligns with our discovery of a high centrality of these regions within the structural brain network, correlating with enhanced music perception. It may imply their crucial function in updating sensory representations for the redirection of attentional focus to relevant items within the stimuli, enhancing their discrimination. Previous research has shown that the microstructural organization of dorsal fronto-parietal white matter pathways, such as the anterior subdivision of the right superior longitudinal fasciculus (SLF I), is related to musical perceptual abilities in non-musicians. Increased white matter coherence in this tract is positively correlated with the speed of musical learning[119]. In both interpretations, higher centrality and communication efficiency of this region can be assumed to aid sensory integration, attentional focus towards pertinent stimuli, and comparison between auditory sequences. The structural organization of the superior parietal lobule may be specialized for the attentional processing and integration of complex sensory inputs, such as those required in music perception, rather than

the more abstract and manipulative cognitive processes involved in WM.

Neurocognitive variability, a hallmark of the human brain, plays a pivotal role in shaping the diverse range of abilities observed across various cognitive and cultural domains. Although extensive research has explored the interplay of brain function, cognitive abilities, and cultural skills, these elements have largely been studied in isolation, leaving a unified theoretical framework elusive. The neuronal recycling hypothesis[120] provides a potential solution, positing that the brain repurposes its older circuits—initially evolved for general cognitive functions—to accommodate evolutionarily more recent cultural skills, all while maintaining their original constraints. Consequently, the spectrum of individual proficiencies within cultural domains is closely linked to the structural and functional nuances of the neural circuits they co-opt as well as to the cognitive functions these circuits originally support[121]. Studies corroborating this theory reveal a significant correlation between general cognitive functions and cultural behavior proficiency, highlighting neural overlap across these domains[122]. Individual network-level constraints in neurocognitive systems may provide a unique neuronal niche through which cultural material is filtered and to which it may eventually adapt. During the cultural transmission of music, minor inter-individual differences in neural information processing can manifest themselves in differences in musical behavior[123]. Amplified and spread through cultural evolutionary mechanisms, minor neurocognitive differences can have large system-level effects, such as diversity within and across human cultures. According to this framework, music can be seen as a useful model system to investigate the link between variations in our innate neurocognitive machinery and large-scale cultural phenomena.

While our study contributes to understanding the perceptual aspects of musicality, musical ability extends far beyond mere perception[9]. The ability to decode emotions from auditory stimuli is another critical component that forecasts musicality[79]. Research has shown a correlation between non-musicians' capacity to recognize emotions expressed through non-verbal vocalizations, facial expressions, and their musical proficiency[124]. Our findings reveal a correlation between melodic and rhythmic perceptual abilities and the subjective experience of emotions in music[10]. This supports the idea of musicality as a multifaceted trait consisting of various interconnected elements where one ability can influence another. However, our findings suggest that these traits might be driven by the activity and organization of partly distinct brain networks[56]. The absence of a significant correlation between self-reported emotional responses to music, WM, and the organizational properties of frontoparietal networks suggests that the neural architecture underlying emotion-related aspects of musical skills may be partially distinct from the ones involved in cognitive and perceptual aspects of music processing. There is compelling evidence suggesting that a large cortical-subcortical brain network subserves music-induced emotions, with the majority of these regions belonging to the limbic and reward system[125]. Future network studies should employ more specific approaches (e.g., between-network connectivity) to understand how trait- or function-specific brain networks interact, giving rise to our capacity for music.

This work presents some limitations. Our research provides information into the relatively static network-based underpinnings of human musicality, which are relevant to understanding the brain network predispositions of musical skills. Yet, further relevant information could be gained by looking at brain dynamics during music listening. Applying dynamic network-based methods on task-based fMRI data[33,34] could reveal the transient states and the network reconfigurations predicting music perceptual abilities. A mechanistic understanding of frontoparietal dynamics in music perception could also assist in the creation of personalized neurostimulation protocols that enhance neural information integration, with implications in music therapeutic interventions and educational settings. Research conducted in the past has demonstrated the potential of transcranial direct current stimulation (tDCS), a noninvasive method of brain stimulation capable of modulating cortical excitability[126]. Specifically, tDCS applied to the right prefrontal region has been found to enhance WM performance even several months following the completion of training[127–129]. This enhancement has been observed across multiple domains of WM[129]. Targeting the right middle frontal gyrus (rMFG) via tDCS could be a strategic approach to enhance music perception capabilities in individuals with music perceptual disabilities[130] through a mediation of WM, a promising avenue for clinical research.

The second limitation pertains to the tentative link we propose between neural variability and cultural phenomena. Our inference is based on limited evidence, primarily due to our participant sample predominantly representing a single European culture[131]. Although recent studies suggest the applicability of the Musical Ear Test (MET) across diverse cultures[53], future research should aim for further cross-cultural validation of the MET to confirm its effectiveness and generalizability[50]. Furthermore, replicating our neural findings with participants from different cultural backgrounds is crucial to robustly support preliminary evidence from macro-cultural studies that associate neurogenetic variability with broad cultural phenomena[85]. Finally, we limited our research to the neurocognitive bases of basic, musical perceptual abilities in music listeners. The MET measures melody and rhythm perceptual skills. However, musical ability is more than mere perception and encompasses a broader range of skills, including emotional capacity, creativity, and social and motor skills[9]. Future work should assess how the network-level processes behind these capacities are integrated, most likely in high-level hubs[132], to provide a more comprehensive understanding of the neurocognitive foundations of human musical ability.

To conclude, our study uses a network science approach to elucidate the complex interrelationship between neurocognitive variability and the spectrum of musical perceptual abilities seen across humans. Our results suggest that intrinsic communication efficiency and integration capacity within FPN core circuitry may aid music perception faculties, with distinct contributions from functional and structural network configurations. The functional topology of right prefrontal regions may facilitate domain-general cognitive functions like WM that support musicality. In contrast, structural properties of superior parietal cortices may subserve sensory or attentional processes more directly tied to auditory capabilities. Overall these findings contribute to elucidating the distinct role of functional and structural neurocognitive variability in support of musical abilities, extending their roots to specific informational properties of brain networks. They provide a framework for future exploration into the neurobiological foundations of human musicality.

## Methods

### Participants

Data were acquired across multiple sessions at Aarhus University and Aarhus University Hospital from healthy individuals as part of the EU COST Action CA18106 The Neural Architecture of Consciousness. The project protocol received ethical approval from De Videnskabsetiske Komitéer for Region Midtjylland, Denmark. The scanning session included the collection of resting-state fMRI, high-angular resolution diffusion imaging (HARDI), and multi-parameter mapping data[133]. Approximately one week prior to undergoing scans, participants completed the Goldsmiths Musical Sophistication Index (Gold-MSI) questionnaire in an online session. Typically within a few weeks of the scans, in an optional session, they completed the Musical Ear Test (MET) and Wechsler Adult Intelligence Scale, Fourth Edition (WAIS-IV). Recruitment of participants was conducted via the Center of Functionally Integrative Neuroscience (CFIN) at Aarhus University, leveraging both the university's participant database and local advertising. A total of 300 adult participants, with no personal history of neurological or psychiatric disorders and no hearing deficits, consented to the study, were financially compensated for their participation, and completed the MRI scanning session (see "MRI acquisition") as well as the Gold-MSI questionnaire (which were both mandatory for study participation).

A subset of these participants ($n = 241$; 135 females, 18–49 years of age) completed the optional MET test session. Biological sex was self-reported by the participants. In terms of musical training, 60% had no music lessons ($n = 145$), 36% had up to 5 years of training ($n = 88$), and only a small subset ($n = 8$) had over 6 years of training and were classified as musicians[134] (Suppl. Fig. 1). This classification is based on Zhang et al. (2018) review of the literature which concluded that 6 years of training represents the general consensus for classifying someone as a musician, even when considering other factors such as the age of onset, intensity of practice, and type of training. Following multivariate outlier analysis, 9 participants were excluded due to significant deviations in Gold-MSI and MET scores, identified via PCA and Euclidean distance criteria (Suppl. Fig. 2). Structural brain network construction failed in 7 more participants. Consequently, the analysis on functional and structural brain networks in relation to MET scores was conducted with 232 (Suppl. Table 1) and 225 participants (Suppl. Table 2), respectively. Additionally, a subset of these participants ($n = 201$) had their domain-general cognitive abilities assessed using the Wechsler Adult Intelligence Scale, Fourth Edition (WAIS-IV)[78]. Only the Working Memory Index (WMI) was used for this study. Graph theory analysis incorporating WMI scores included 201 participants for functional (Suppl. Table 3) and 195 for structural brain network analyzes (Suppl. Table 4).

### Musical abilities

**Musical Ear Test (MET).** The Musical Ear Test (MET) comprises 104 trials: 52 melodic phrase pairs in the Melody subtest and 52 rhythmic phrase pairs in the Rhythm subtest. Before testing began, participants were instructed to use headphones and minimize distractions. Participants evaluated whether sequences in each trial − piano tones for Melody and drum beats for Rhythm − were identical, with deviations involving at least one tone (Melody) or inter-onset interval (Rhythm). Feedback was restricted to initial practice trials. Inter-trial intervals in the audio were capped at 1500 ms for Melody and range from 1659 to 3230 ms for Rhythm, thus standardizing MET duration. Scoring awards one point for each correct response, with Melody and Rhythm subtest scores each calculated as the percentage of correct answers out of 52. The percentage of MET total scores was calculated as the percentage of correct answers out of the sum of these subtest scores (i.e., 104). Percentage of MET total scores, strongly correlating with melodic ($r = 0.89$) and rhythmic scores ($r = 0.85$) (Suppl. Figure 3), was the primary metric in subsequent analyzes, representing musical competence.

**Goldsmiths Musical Sophistication Index (Gold-MSI).** The Gold-MSI is a 38-item self-report questionnaire assessing musical behaviors, experiences, and skills. It comprises five subscales: Active Engagement (9 items, e.g., daily attentive music listening duration), Perceptual

Abilities (9 items, e.g., identifying out-of-tune singing or playing), Music Training (7 items, e.g., years of formal music theory training), Singing Abilities (7 items, e.g., accuracy in matching recorded notes while singing), and Emotions (6 items, e.g., selecting music for mood enhancement). Additionally, a General Factor score is derived from 18 representative items across these subscales. For the first 31 items, responses are rated on a 7-point Likert scale, ranging from complete disagreement to complete agreement. The last eight items feature variable response options. Scores for the subscales of the Gold-MSI were calculated with the GMSI Scorer App (https://shiny.gold-msi.org/gmsiscorer).

### Working memory abilities

**Wechsler adult intelligence scale.** In the Wechsler Adult Intelligence Scale, Fourth Edition (WAIS-IV), our analysis focused solely on the Digit Span and Arithmetic subtests to assess WM in participants. The Digit Span subtest comprises three distinct tasks: Digit Span Forwards, Digit Span Backwards, and Digit Span Sequencing. The Digit Span Forwards task involves the oral presentation of number sequences by the experimenter, which participants are required to replicate verbatim. Performance is measured by the number of sequences accurately recalled. In the Digit Span Backwards task, participants must reverse and repeat the sequences, with scores reflecting the count of sequences correctly reproduced in reverse order. The Digit Sequencing task necessitates rearranging spoken numbers in ascending order, scored based on the number of sequences correctly ordered. The Arithmetic subtest involves mentally solving arithmetic problems presented verbally as their difficulty and memory load increases. For each subtest, raw scores were normalized to age-corrected z scores, with a mean of zero and standard deviation of one, wherein higher scores signify enhanced performance. The Working Memory Index (WMI) is derived by summing the scaled scores from these tasks and subsequently converting this aggregate into an index score using a standard conversion table.

### MRI acquisition

Data were acquired using a Siemens Magnetom Prisma-fit 3 T MRI scanner. Following an initial scout scan, two resting-state fMRI sequences (12 and 6 minutes) were run, accompanied by quantitative multi-parameter mapping[133] (around 20 minutes) −used here for synthetically generated T1-weighted images− and high-angular resolution diffusion imaging (HARDI) (around 10 minutes), within a one-hour scanning session. For each participant, 1500 functional volumes were acquired (TR, 700 ms; TE, 30 ms; voxel size 2.5 mm$^3$).

The MPM protocol was implemented based on the Siemens vendor sequence. Three-dimensional (3D) data acquisition consisted of three multi-echo spoiled gradient echo scans (i.e. fast low angle shot [FLASH] sequences with MT, T1, and PD contrast weighting). Additional reference radio-frequency (RF) scans were acquired. The acquisition protocol had the following parameters: TR of PD- and T1-weighted contrasts: 18 ms; TR of MT-weighted contrast: 37 ms; minimum/maximum TE of PD-, T1- and MT-weighted contrasts: 2.46/14.76 ms; flip angles for MT-, PD- and T1-weighted contrasts: 6°, 4°, 25°, respectively; six equidistant echoes; 1 mm isotropic reconstruction voxel size; Field of view 224′ 256′ 176 mm; AP phase encoding direction; GRAPPA parallel imaging speedup factor of 2; T1w, PDw and MTw acquisition times: 3:50, 3.50, 7.52. The acquisition of low-resolution 3D spoiled gradient echo volumes was executed using both the RF head coil and the body coil. This dual acquisition facilitated the generation of a relative net RF receive field sensitivity (B1 −) map for the head coil[135–137]. The approach obtained rapid acquisition by maintaining a low isotropic spatial resolution of 4^3 mm^3, a short echo time (TE) approx 2 ms, and a reduced flip angle of 6°, avoiding the use of parallel imaging acceleration or partial Fourier. This procedure of capturing volume pairs with the head and body coils was systematically repeated prior to the acquisition of each of the MT, PD, and T1 contrasts.

The sequence used to collect HARDI images included: 75 diffusion directions at b = 2500 s/mm$^2$; 60 directions at b = 1500 s/mm$^2$; 21 directions at b = 1200 s/mm$^2$; 30 directions at b = 1000 s/mm$^2$; 15 directions at b = 700/mm$^2$; 10 directions at b = 5 s/mm$^2$, with the different b-shells acquired in the same series (flip angle = 90◦, TR/TE = 2850/71 ms; voxel size = 2 mm$^3$; matrix size = 100 x 100, number of slices = 84). The phase-encoding direction was anterior to posterior (AP). An opposite phase-encoding direction (PA) was also acquired (b = 0 s/mm$^2$) to allow EPI distortion correction[138].

### Neuroanatomical data processing

Synthetic T1-weighted images were generated using the longitudinal relaxation rate (R1) and effective proton density (PD) high resolution maps (acquired during the MPM sequence protocol). First, both maps were thresholded in order to achieve the required FreeSurfer units. The R1 map was converted to a T1 map by taking its reciprocal and thresholded at zero. This was scaled by a factor of 1000. The PD map was thresholded by zero and scaled by 100. All manipulations were performed using FSL maths commands. Subsequently, the "mri_synthesize" FreeSurfer command was applied to create a synthetic FLASH image based on previously calculated T1 (thresholded R1 map) and proton density map. The optional flagged argument for optimal gray and white matter contrast weighting was used with the following parameters 20, 30, and 2.5. Finally, the synthetic T1-weighted image was divided by four to achieve the scale that FreeSurfer expects.

The synthetic T1-weighted image was preprocessed using fMRI-Prep 21.0.2[139] (RRID:SCR_016216), which is based on Nipype 1.6.1[140] (RRID:SCR_002502). The T1-weighted image were corrected for intensity non-uniformity (INU) using the N4BiasFieldCorrection[141], part of the ANTs 2.3.3[142] (RRID:SCR_004757). This corrected image served as the T1w-reference throughout the preprocessing workflow. Skull stripping was performed on this reference image using a Nipype implementation of the antsBrainExtraction.sh workflow (from ANTs), with the OASIS30ANTs as the target template. Brain tissue segmentation of cerebrospinal fluid (CSF), white-matter (WM) and gray-matter (GM) was performed on the brain-extracted T1w using fast[143] (FSL 6.0.5.1; RRID:SCR_002823). Brain surface reconstruction was carried out using FreeSurfer's recon-all function[144] (version 6.0.1; RRID:SCR_001847), and the brain mask estimated previously was refined with a custom variation of the method to reconcile ANTs-derived and FreeSurfer-derived segmentations of the cortical gray-matter of Mindboggle[145] (RRID:SCR_002438). Volume-based spatial normalization of the brain images to the two standard spaces (MNI152NLin2009cAsym, MNI152NLin6Asym) was executed through nonlinear registration with antsRegistration (ANTs 2.3.3), using brain-extracted versions of the T1w reference and the T1w template. The templates employed for this normalization included the ICBM 152 Nonlinear Asymmetrical template version 2009c (RRID:SCR_008796; TemplateFlow ID: MNI152NLin2009cAsym) and FSL's MNI ICBM 152 non-linear 6th Generation Asymmetric Average Brain Stereotaxic Registration Model[146] (RRID:SCR_002823; TemplateFlow ID: MNI152NLin6Asym).

### dMRI processing and structural brain network construction

The diffusion MRI (dMRI) data was preprocessed using custom MATLAB scripts developed internally at the Center of Functionally Integrative Neuroscience (CFIN). The preprocessing steps included noise reduction adapted from the approach by Veraart et al.[147], correction of Gibbs ringing artifacts following the method described by Kellner et al.[148], and motion, eddy currents, and field distortion corrections using the top-up and eddy tools from the FSL toolbox[149]. The generation of structural brain networks was performed using the MRtrix3 software toolkit. The analysis involved several steps per participant. We first created a 5-tissue-type (5tt) image which contained masks of different tissue types (cortical gray

matter, deep gray matter, white matter, CSF and "other") within the brain and is essential for Anatomically-Constrained Tractography (ACT). Co-registration was then performed to align T1-weighted and DWI images. A response function was created for each major tissue type (white matter, gray matter, cerebrospinal fluid) for each participant. The individual participant response functions were used to create group-level response functions. Multi-Shell Multi-Tissue Constrained Spherical Deconvolution (MSMT-CSD), was used to estimate Fiber Orientation Distributions (FODs) within each voxel of the brain, followed by normalization.

Next, whole-brain probabilistic tractography was performed using the ACT framework and backtracking. The maximum attempted number of streamlines was $1*10^9$ streamlines with 10 million streamlines per brain network being selected. Each seed was determined dynamically from the FOD image using the SIFT model. The FOD cutoff was 0.06, the maximum length of each selected streamlines was 250 mm while the minimum was 20 mm. The SIFT2 model was then applied to the data. Structural networks were then generated using the Destrieux parcellation for the cortex and the FSL FIRST segmentations for the subcortical structures. Each network was multiplied by the SIFT proportionality coefficient (mu). Finally, a custom automated pipeline for visualizing the diffusion data in a structured and standardized way was run for quality control. We generated jpegs of the 5TT images alongside GIFs of the registration of the T1w image to the B = 0 image. These visualizations were used to ensure that the processing pipeline worked correctly.

### rsfMRI processing and functional brain network construction

The processing of resting-state (rs-fMRI) volumes was implemented by using default surface-based preprocessing routines from the SPM CONN toolbox (Whitfield-Gabrieli (http://www.nitrc.org/projects/conn)[150], implemented in Matlab (2016b). Functional data was realigned and unwarped without field maps using SPM12 (r7487), employing a 6-parameter transformation for alignment and b-spline interpolation for resampling. Outliers were identified using ART[151] based on framewise displacement and global BOLD signal deviations, and an average reference BOLD image was created for each participant excluding all outlier volumes. Coregistration of functional and anatomical data was achieved using mutual information. Functional images were then mapped onto the cortical surface, averaging data across layers between the pial and white matter surfaces. Finally, surface-level functional data were smoothed using 40 iterative diffusion steps. Our denoising process involved a standard pipeline, regressing out confounds like white matter and cerebrospinal fluid (CSF) signals, motion artifacts, outlier scans, session effects, and linear trends. This included the use of CompCor for noise component extraction from white matter and CSF. Bandpass frequency filtering was applied to the BOLD timeseries to retain frequencies between 0.008 Hz and 0.09 Hz. The effective degrees of freedom of the BOLD signal post-denoising were estimated for all participants.

We estimated region-to-region connectivity matrices across 16 regions of interest (ROIs) by calculating the functional connectivity strength (Fig. 2). This was represented by Fisher-transformed bivariate correlation coefficients derived from a weighted general linear model (GLM) and stored in a functional connectivity matrix. The GLM accounted for associations between BOLD signal timeseries of ROI pairs, with weighting to mitigate transient magnetization effects at the start of each run. The connectivity matrix only included Destrieux's cortical nodes (148x148). From this matrix, we selected 16 nodes for one frontoparietal network of interest and one occipital control network (16x16) (Fig. 2).

### Graph theory analyzes

Graph theory analyzes for functional and structural connectivity matrices, and for both frontoparietal and occipital networks, followed the same pipeline. Connectivity matrices (16×16) were thresholded at a fixed network-level cost range (k) (0<k<1), resulting in binarized, undirected adjacency matrices. The analysis incorporated both positive and negative rs-FC values. To avoid reliance on specific and arbitrary threshold values (e.g., k = 0.15)[152], graph metrics were aggregated across multiple thresholds (k = 0.15–0.30, interval 0.01)[153]. Within this range, brain networks show small-world features (GE and LE have, respectively, larger values than lattice and random graphs of equal size and cost values; Supplementary Figs. 3–4). From the matrices, four node-level graph theory metrics were computed using the Brain Connectivity Toolbox (BCT)[88]: clustering coefficient, local efficiency, global efficiency, and betweenness centrality (Fig. 3). All these metrics offer clear interpretability and are prevalent in network studies. A second-level General Linear Model (GLM) included MET as the main predictor and age, sex, and musical training (Gold-MSI questionnaire) as nuisance regressors. Node-level $p$-values were adjusted for multiple comparisons using a false discovery rate of q < 0.05 (two-tailed), for each graph metric.

### Reporting summary

Further information on research design is available in the Nature Portfolio Reporting Summary linked to this article.

## Data availability

Data cannot be shared publicly as it is part of an ongoing study, and thus considered unanonymized under Danish law, even if pseudonymized. Researchers who wish to access the data may contact Dr Kristian Sandberg (kristian.sandberg@cfin.au.dk) at The Center of Functionally Integrative Neuroscience and/or The Technology Transfer Office (TTO@au.dk) at Aarhus University, Denmark, to establish a data sharing agreement. After permission has been given by the relevant ethics committee, data will be made available to the researchers for replication purposes. As the project is ongoing, sharing requests for other purposes will be evaluated on a case-by-case basis.

## Code availability

Code for reproducing graph theory results is available on GitHub: https://github.com/MassimoLumaca/neuroMET. Code for reproducing diffusion analysis is available on GitHub: https://github.com/MassimoLumaca/neuroARC.

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

## Acknowledgements

This article is based upon work from COST Action CA18106 (The Neural Architecture of Consciousness), supported by COST (European Cooperation in Science and Technology). We thank Signe Kirk Brødbæk, Simon Durand, Nina Dyrberg, Sara Kolding, Audrey Mazancieux, Dunja Paunovic, Bianka Rumi, and Povilas Tarailis for their assistance in data collection. We thank Jelle van der Werff for technical helps. The Center for Music in the Brain (MIB) is funded by the Danish National Research Foundation (project number DNRF117). C.J.B. was funded by the Measuring the Architecture of Consciousness (MARC) Project, financed by the Malta Council for Science & Technology (MCST) through the Research Excellence Programme (REP-2022-005), for and on behalf of the Foundation for Science and Technology.

## Author contributions

M.L. conceived the hypothesis. K.S. designed the study. K.S. recruited the resources for the experiment. K.S., M.A.M., J.H.H., and K.V. collected the data. N.J. digitalized the responses to the musical questionnaires. M.L. performed pre-processing, denoising, and connectome construction of resting-state functional data. K.S. performed pre-processing of diffusion data. C.J.B performed processing and connectome construction of diffusion data. M.L. performed graph theory and mediation analyzes. M.L. prepared the figures. P.K., G.B., and A.R. provided essential help to frame the introduction within the neuroscientific literature of musical abilities. M.L. wrote the first draft of the manuscript. C.J.B and K.S. wrote a significant part of the methods. G.B., K.S., P.K., P.V., V.P.N., and C.J.B., provided useful comments and edits during the revision of the manuscript. All the authors contributed to and approved the final version of the manuscript.

## Competing interests

The authors declare no competing interests.
