## [Peer Review File · Nature Communications]

Frontoparietal network topology as a neural marker of musical perceptual abilitiesEditorial Note: This manuscript has been previously reviewed at another journal that is not operating a transparent peer review scheme. This document only contains reviewer comments and rebuttal letters for versions considered at *Nature Communications*.

Reviewer #1 (Remarks to the Author):

The authors have diligently and appropriately addressed my initial concerns, as well as, in my opinion, those of the other reviewers. The paper is stronger and clearer as a result. I have no further comments and would support acceptance in its current state.

Reviewer #2 (Remarks to the Author):

I do not have any remaining concerns about this work or any additional comments for further improvement of the manuscript; the revised manuscript is significantly improved, well-written, and provides an innovative approach and novel evidence for understanding the brain bases of human musicality. Indeed, this study may guide future research on the neural bases of human musicality, promoting a graph theoretical approach for assessing complex brain network properties of both functional and structural networks in relation to cognitive (or affective) processes underlying aspects of musicality.

Reviewer #3 (Remarks to the Author):

I reviewed a previous version of this manuscript and I find this version of the manuscript much improved based on the authors' incorporation of feedback from all 3 reviewers. The findings are novel and innovative and I believe they add significantly to the literature on domain-general skills, musical ability, and neural underpinnings.

I particularly appreciate the tempering of the cultural claims and a shift in focus instead on individual differences. Further, the new figure 3 is an extremely helpful reference point for interpreting the results throughout the rest of the paper.

The addition of the non-significant relationships in Figure 4 is also helpful for understanding that all the n.s. findings were in the same direction as other regions, they just did not reach significance. The supplementary material is also very helpful for allowing readers to inspect the correlations for all regions (esp. occipital regions) on their own.

Finally, I think the authors have done a very nice job adding nuanced perspectives about the relationships between musical perceptual abilities and musicality, individual differences and culture, and emotions and working memory networks.

Response to reviewers

Frontoparietal network topology as a neural marker of musical perceptual abilities

Reviewer expertise:

Reviewer #1: Network neuroscience, graph theory, creativity

Reviewer #2: Music perception, creativity

Reviewer #3: Music perception, auditory cognition

Reviewers' Comments:

Reviewer #1:

The authors have diligently and appropriately addressed my initial concerns, as well as, in my opinion, those of the other reviewers. The paper is stronger and clearer as a result. I have no further comments and would support acceptance in its current state.

Reviewer #2:

I do not have any remaining concerns about this work or any additional comments for further improvement of the manuscript; the revised manuscript is significantly improved, well-written, and provides an innovative approach and novel evidence for understanding the brain bases of human musicality. Indeed, this study may guide future research on the neural bases of human musicality, promoting a graph theoretical approach for assessing complex brain network properties of both functional and structural networks in relation to cognitive (or affective) processes underlying aspects of musicality.

Reviewer #3:

I reviewed a previous version of this manuscript and I find this version of the manuscript much improved based on the authors' incorporation of feedback from all 3 reviewers. The findings are novel and innovative and I believe they add significantly to the literature on domain-general skills, musical ability, and neural underpinnings.

I particularly appreciate the tempering of the cultural claims and a shift in focus instead on individual differences. Further, the new figure 3 is an extremely helpful reference point for interpreting the results throughout the rest of the paper.

The addition of the non-significant relationships in Figure 4 is also helpful for understanding that all the n.s. findings were in the same direction as other regions, they just did not reach significance. The supplementary material is also very helpful for allowing readers to inspect the correlations for all regions (esp. occipital regions) on their own.

Finally, I think the authors have done a very nice job adding nuanced perspectives about the relationships between musical perceptual abilities and musicality, individual differences and culture, and emotions and working memory networks.

We thank all the reviewers for their positive feedback and constructive comments. We have made minor adjustments to the title based on the previous revision, changing "neuromarker" to "neural marker" and "music perceptual" to "musical perceptual" for improved clarity and precision.

In response to Reviewer 2's request for a supplementary analysis, we discovered and corrected a small bug in the code for importing Emotion's Gold-MSI data. Consequently, we have updated Supplementary Figure 3 (last row) and Supplementary Tables 13-16. These changes are minor, yield comparable statistical results, and do not affect any of the original conclusions on Gold-MSI's Emotion data.